# Prior Advanced Care Planning and Outcomes of Cardiopulmonary Resuscitation in the Emergency Department of a Comprehensive Cancer Center

**DOI:** 10.3390/cancers16162835

**Published:** 2024-08-13

**Authors:** Adriana H. Wechsler, Marcelo Sandoval, Jayne Viets-Upchurch, Maria Cruz Carreras, Valda D. Page, Ahmed Elsayem, Aiham Qdaisat, Sai-Ching J. Yeung

**Affiliations:** Department of Emergency Medicine, The University of Texas MD Anderson Cancer Center, Houston, TX 77030, USA; msandoval@mdanderson.org (M.S.); jviets@mdanderson.org (J.V.-U.); mcruz3@mdanderson.org (M.C.C.); vpage@mdanderson.org (V.D.P.); aelsayem@mdanderson.org (A.E.); aqdaisat@mdanderson.org (A.Q.); syeung@mdanderson.org (S.-C.J.Y.)

**Keywords:** cardiopulmonary resuscitation, cancer, emergency department, advanced care planning, goals of care, do not resuscitate, mortality

## Abstract

**Simple Summary:**

As more cancer patients with advanced disease present to the emergency department (ED), data on outcomes of cardiopulmonary resuscitation (CPR) are needed to help counsel patients before and during an acute event. We investigated the characteristics of cancer patients who required CPR, their outcomes, and how prior advanced care planning (ACP) influenced these outcomes. Few studies have specifically looked at these data in an emergency department, where medical history is limited and the need for acute intervention often precludes discussion of therapeutic limitations. We found cardiopulmonary resuscitation of cancer patients to be rare in the ED. Although the return of spontaneous circulation (ROSC) is often attained, very few patients survive to discharge. Patient characteristics, resuscitation success, overall mortality, and the cost of care did not differ between patients with and without ACP. However, patients with ACP had shorter hospital and intensive care unit (ICU) stays and higher rates of conversion to do-not-resuscitate (DNR) status post-resuscitation.

**Abstract:**

Cardiopulmonary resuscitation (CPR) outcomes vary for patients with cancer. Here, we characterized cancer patients who underwent CPR in the emergency department (ED), their outcomes, and the effects of advanced care planning (ACP). The hospital databases and electronic medical records of cancer patients at a comprehensive cancer center who underwent CPR in the ED from 6 March 2016 to 31 December 2022 were reviewed for patient characteristics, return of spontaneous circulation (ROSC), conversion to do-not-resuscitate (DNR) status afterward, hospital and intensive care unit (ICU) length of stay, mortality, cost of hospitalization, and prior GOC discussions. CPR occurred in 0.05% of all ED visits. Of the 100 included patients, 67 patients achieved ROSC, with 15% surviving to hospital discharge. The median survival was 26 h, and the 30-day mortality rate was 89%. Patients with and without prior ACP had no significant differences in demographics, metastatic involvement, achievement of ROSC, or in-hospital mortality, but patients with ACP were more likely to change their code status to DNR and had shorter stays in the ICU or hospital. In conclusion, few cancer patients undergo CPR in the ED. Whether this results from an increase in terminally ill patients choosing DNR status requires further study. ACP was associated with increased conversion to DNR after resuscitation and decreased hospital or ICU stays without an increase in overall mortality.

## 1. Introduction

As the incidence and survival rates for cancer continue to rise worldwide, the number of cancer patients presenting to the emergency department (ED) has also increased. Whereas morbidity and mortality rates have improved for many patients, those needing emergency care often have advanced disease, and a palliative approach may be more appropriate [1]. Discriminating patients likely to benefit from aggressive resuscitation from those who will not is becoming increasingly important. In the ED, cancer patients with incomplete or unknown histories will often present requiring urgent life-saving interventions, which limits goal-setting discussions, leading to inappropriate or unintended resuscitation [2].

Successful cardiopulmonary resuscitation (CPR) with meaningful recovery is limited in the general population (with one meta-analysis by Zhu from 2015 demonstrating a rate of 15%), with similar rates for cancer subgroups (2–18%) [3,4,5,6,7]. Despite these lackluster rates, large cardiovascular care organizations are reluctant to endorse any pre-cardiac arrest prediction tools to identify patients who are unlikely to survive with favorable neurologic outcomes and help guide goals of care discussions [8].

In the decade since the Zhu data, there has been an increasing emphasis on establishing realistic goals of care for patients with advanced cancer and prioritizing their quality of life. Cost containment and resource constraints, such as those experienced during the COVID-19 pandemic, have accelerated this agenda [9,10,11]. For example, the rise in seriously ill patients during the COVID-19 pandemic led to a concurrent rise in palliative care offerings, even in EDs [12]. Advanced care planning (ACP) became more prevalent without evidence of a decrease in overall survival [9,10,13] Nevertheless, advanced care planning in many cancer patients presenting to the ED remains inadequate, despite the well-documented need for establishing goals of care (GOC) before an urgent medical crisis [1,14,15,16]. One study demonstrated that only 24% of hospitalized cancer patients discussed end-of-life care with any provider despite 82.5% wanting to do so [16].

Leaders in both emergency and palliative medicine have called for the introduction of palliative care in EDs [17,18]. In 2015, the National Institutes of Health created an oncology-specific emergency and urgent care consortium, which identified ED-based palliative care as a priority [19]. From 2020 through 2023, an alliance of 10 dedicated cancer centers issued a call to improve goal-concordant care rooted in national quality initiatives for seriously ill patients, such as communication skills training, structured electronic record documentation of ACP, mandated GOC conversations, and tracking of metrics [20].

Over the past 4 years, our institution responded to these recommendations by launching a *Goals of Care Project* to identify populations at risk for decompensation and to establish GOC early in their trajectory [10]. The project resulted in decreased ICU admissions, hospital length of stay, and mortality, alongside improved symptom scores and performance status [9,10]. Yet, how this focus on palliative care impacted the incidence and outcomes of cancer patient resuscitation in the ED has not been studied.

The aim of our study was to update the data on CPR outcomes for cancer patients presenting to the ED and to explore the statistical association of prior GOC discussion or ACP with the post-resuscitation course. Our study focuses on patients in the ED, where the urgency of resuscitation may supersede obtaining a full medical history or engaging with the patient in discussions of prognosis, unlike the hospital ward or ICU. Some of our patients made their resuscitation wishes known through prior directives or discussions in the ED, but others never had this opportunity prior to CPR. We hypothesized that the patients who had prior goals of care discussions and were willing to undergo resuscitation did so because they had a more favorable cancer prognosis and, therefore, we would expect better outcomes.

## 2. Materials and Methods

We conducted a single-center, retrospective observational study using chart review and collection of data from our institutional databases for all patients with an identified malignancy greater than 18 years of age who underwent CPR from 6 March 2016 to 31 December 2022 in the ED of our National Comprehensive Cancer Network-designated cancer center.

We obtained patient charts from mandatory code-blue reporting within our quality and safety recording system for patients who underwent CPR from 6 March 2016 to 30 October 2022 and merged them with those reported separately by our institutional code-blue committee from 1 September 2019 to 31 December 2022. We chose 6 March 2016 as the start date because it coincided with the initiation of our new institutional electronic medical record system (EPIC).

Patients were included in our study cohort if they had an identified cancer, were at least 18 years old, and received advanced cardiovascular life support (ACLS) in the ED or had it initiated in the field by emergency medical personnel and then continued in the ED. We included chest compressions, electrical cardioversion, or both in our definition of ACLS. Patients were excluded if CPR was halted by the emergency medical personnel prior to arrival to the ED, if CPR was not resumed owing to ROSC, or if they were declared dead on arrival to the ED. Patients who experienced cardiac arrest outside the ED, such as on a medical floor or in the ICU, were also excluded. The study was approved by The University of Texas MD Anderson Cancer Center Institutional Review Board, which waived the requirement for informed consent for this retrospective study.

Data were abstracted from the electronic medical record using standardized questions and collected on a research electronic data capture platform (REDcap) hosted at MD Anderson [21]. Variables were given strict definitions, and the researchers were trained to follow consistent methods of data abstraction. To compare patients who had prior GOC discussions or ACP with those who did not, we methodically searched for documentation of GOC discussions, ACP, or “code” or “DNR status” conversations in provider notes over the year before the code event and in all entries in the dedicated ACP section of the medical record chart. If no such documentation was found, we concluded that prior GOC or ACP did not occur.

The REDcap data were then reviewed by another one of the investigators to ensure adherence to protocols and definitions of variables. All discrepancies in data collection were adjudicated by the first investigator. Institutional databases were used to gather visit-related variables, patient demographics, Charlson Comorbidity Index (CCI) values, hospitalization costs, length-of-stay data, and mortality outcomes. Descriptive statistics were used to analyze and report the data, with continuous variables reported as means and standard deviations (SDs) or medians and interquartile range (IQRs) as appropriate, whereas categorical variables were reported as frequencies and percentages. Significance was appraised using chi-square tests, the Fisher exact test, Student’s *t*-test, or nonparametric tests (i.e., the Wilcoxon–Mann–Whitney test) when normality assumptions were not met. The normality of the continuous variables was assessed using histograms, box plots, Q-Q plots, and the Shapiro–Wilk test, for which only age met the normality assumption. Survival rates were estimated using Kaplan–Meier survival analysis for the whole cohort followed by a log-rank test to estimate the difference in overall survival between patients with and without ACP notes or GOC documentation prior to the resuscitation event. All statistical analyses were conducted using the R computing language (Version 4.3.1), in which a two-sided test with a *p* value of up to 0.05 was considered significant.

## 3. Results

### 3.1. Patient Demographics and Incidence of Resuscitation

Over the study period, there were 183,483 ED visits involving 74,896 unique patients with only 100 patients (0.13%) meeting the inclusion criteria (≥18 years old, established cancer, CPR initiated before arrival to or while in the ED, and underwent resuscitative efforts in the ED) (Figure 1). A resuscitation event occurred in 100 of 183,483 (0.05%) of all ED visits or 0.55 of every 1000 patient visits and in 0.13% of unique patients.

The patients had a mean age of 62 years and were equally distributed by sex. Overall, 63% of the patients were White, 19% were Black, and 16% were Hispanic ethnicity, regardless of race. The median CCI for all resuscitated patients was 6 (Table 1).

Most patients had gastrointestinal cancers (17%), followed by lung neoplasms (16%), leukemia (14%), head and neck cancers (13%), other hematologic cancers (11%), genitourinary cancers (9%), gynecologic cancers (7%), and breast cancers (4%). Distant metastases were present in 66% of the patients without hematologic or central nervous system cancers. Most patients had received some type of cancer therapy, including chemotherapy, immunotherapy, radiation therapy, targeted therapy, and surgery, within the 2 months before their CPR event (Table 2).

All patients had either chosen to undergo resuscitation or were unable to specify their preferences immediately before CPR. Six patients revoked their prior DNR orders (Table 3).

### 3.2. Outcomes

Of the 100 patients who underwent resuscitation, 67% achieved ROSC and 15% survived to discharge from the hospital (Table 3). The Kaplan–Meier survival curve for the patients who were successfully resuscitated in the ED showed a median survival of 26 h (Figure 2).

Within 24 h, 65% of the patients who underwent CPR were dead (33 never achieved ROSC and 32 died within 24 h of the CPR event). At 48 h post-resuscitation, 72% of the patients had died, and after 1 week, 81% of patients had died. The 30-day, 60-day, and 3-month mortality rates were 89%, 93%, and 94%, respectively (Table 3). Patients with prior ACP achieved ROSC at nearly the same rate as those without ACP (70% vs. 65%) (Appendix A).

We performed a separate analysis in which we expressed the numbers of resuscitation events and ROSC events as percentages of the yearly ED census rates from 2016 to 2022. The yearly number of patients who underwent CPR was too small to identify significant trends over the 7-year study period, but we noticed a small rise in the percentage of patients undergoing resuscitation with an accompanying decline in the fraction achieving ROSC over the last 3 years of the study period. Over the 7-year study period, the patients who underwent CPR accrued more than $7.6 million in hospital costs (Table 4).

### 3.3. Associations with ACP

Overall, 44% of the patients undergoing resuscitation had documentation of GOC discussions, ACP, or resuscitation preferences documented prior to the ED visit (Table 3). The demographics and clinical characteristics stratified by ACP or GOC discussions prior to the resuscitation event are summarized in Appendix A. There was no significant difference in achieving ROSC between patients with or without ACP or GOC discussions prior to the resuscitation event (70.5% vs. 64.3%, respectively; *p* = 0.515; Appendix A). A total of 48 (72%) out of the 67 patients with ROSC changed their code status to DNR after the CPR event (Table 5).

In general, we observed little statistically significant difference in patient demographics, cancer stage, or treatment between the patients with and without documented ACP for the whole cohort (Appendix A), nor for the smaller group that achieved ROSC (Table 5). Women were significantly more likely than men to have documented GOC discussions prior to cardiac arrest (28/50 vs. 16/50; *p* = 0.016) (Appendix A). This difference was also present, albeit insignificantly, in the ROSC group (19/35 vs. 12/32; *p* = 0.169) (Table 5). Patients with ROSC and documented ACP had a higher CCI score than those without ACP (8 vs. 6, *p* = 0.013). Lengths of stay in the ICU and hospital were markedly shorter for patients with documented ACP than those without (ICU, 0.92 days vs. 2.25 days; hospital, 1.17 days vs. 2.6 days). (Table 5). Significantly more patients with prior ACP or goals of care discussion changed their code status to DNR post-resuscitation than those without such documentation (84% vs. 61%; *p* = 0.039), yet in-hospital mortality was nearly equal amongst the two groups (78% vs. 84%; *p* = 0.529) (Table 5). The Kaplan–Meier survival curves for patients with ROSC likewise did not demonstrate a significant difference in survival for those with and without ACP (Figure 3).

## 4. Discussion

This retrospective observational cohort study of cancer patients who underwent CPR in the ED of a dedicated cancer hospital demonstrated that resuscitation was exceedingly rare over the past 7 years (2016–2022), occurring in 0.05% of all ED visits. We found few direct comparisons for this cohort in the literature. A review of all in-hospital (non-prehospital) CPR events in 300–400 U.S. general EDs from 2010 to 2018 based on data from the National Center for Health Statistics showed an incidence rate of 1.2 per 1000 visits, more than twice our cancer hospital’s rate of 0.55 resuscitations per 1000 visits, including our two out-of-hospital cardiac arrests [22]. Further investigation is necessary to understand if this lower incidence of CPR in our ED is a result of more morbidly ill cancer patients choosing DNR status through prior ACP and thus not coming to the ED or not undergoing resuscitation once there. Determining whether this low incidence of CPR is true for patients with cancer in general ED settings also requires further study.

The outcomes of CPR are more frequently studied than its incidence. Although more than two-thirds (67%) of our patients initially achieved ROSC and more than half (59%) made it out of the ED, very few (15%) survived to discharge. Our ROSC rates are higher and survival rates are nearly equal to those in the National Center for Health Statistics study of CPR in general EDs, in which half of the patients died in the ED and the majority of those admitted died in the hospital [22]. However, a true comparison of our small cohort to this national sample is difficult. A study of CPR specific to cancer patients at a large Canadian hospital, but not limited to the emergency department, also demonstrated higher ROSC rates (61%) for patients with cancer compared to the general population and a low in-hospital survival rate (12%) [6].

Most previous studies of resuscitation outcomes in cancer patients focused on inpatients on medical–surgical floors or in critical care units, where, unlike in the ED, a patient’s cancer history and current health condition are available to guide decision-making [23,24,25,26,27]. Such a study was performed at our institution from 2011 to 2015, looking at CPR in cancer patients hospitalized on a floor or in the ICU. The authors noted an even higher rate of ROSC (80% vs. 67%) than in our ED-based study, but their survival-to-discharge rate was as poor as ours (7% for patients with hematologic cancers and 13% for patients with solid cancers vs. 15%) despite fewer of their patients converting to DNR status (34% vs. 80%) and presumably receiving more aggressive critical care [27].

Only 44% of our patients who underwent CPR in the ED had documented GOC discussions, although most had metastatic disease (66% of patients with nonhematologic cancers) and high CCI scores (Table 2 and Table 3). This documentation rate is similar to those in recent studies [28]. Nearly 80% of our patients who achieved ROSC changed their DNR status post-resuscitation, leading us to question if more of these traumatic and expensive events could have been prevented or been better aligned with patient wishes through improved ACP and documentation. We indirectly explored this question by looking at how the effects of prior ACP impacted the clinical course of patients undergoing CPR. When comparing post-resuscitation mortality in patients with and without documented prior GOC discussions, we showed that while the in-hospital mortality rates were equally high in the two groups (84% for patients with ACP or GOC documentation vs. 78% for those without such documentation), those with documented GOC conversations had statistically significant shorter ICU and hospital lengths of stay (0.92 days vs. 2.25 days and 1.17 days vs. 2.6 days, respectively). This may be associated with the fact that more patients with prior GOC discussions transitioned to DNR status following resuscitation. Female patients were considerably more likely to have documented GOC conversations prior to cardiac arrest, which is in keeping with previous studies demonstrating that women are more likely to engage in end-of-life discussions and have more realistic comprehension of their prognoses than men [29].

We hypothesize that prior discussions of care goals enhanced the understanding of their disease and its prognosis, helping patients and their families more quickly transition to DNR status post-arrest. Those who actively participated in ACP discussions may prioritize quality of life over prolonging it through invasive interventions. Pre-existing co-morbidity, reflected in higher CCI scores in those with prior ACP, also may have contributed to their higher rates of early goal-setting conversations, earlier deaths, and, by extension, shorter ICU and hospital lengths of stay. Undoubtedly, the trauma and consequences of resuscitation prompted the transition to DNR status in both groups regardless of prior discussions, but the effect was significantly more pronounced for those having documented GOC discussions than for those who did not (26/31 vs. 22/36; *p* = 0.039), perhaps because they were more prepared to let go after a catastrophic event.

Several studies have outlined the poor outcomes in patients with cancer who undergo CPR. These studies petition for increased and earlier GOC discussions and palliative care in the ED or try to identify characteristics that prognosticate survival [6,26,27,30,31,32]. The most recent such study retrospectively reviewed the cancer type, stage, treatment, and precipitating illness in an ICU cohort of patients with cardiac arrest to try to identify predictors of favorable outcomes [32]. Unfortunately, much of this background information is unavailable when patients are first presenting to the ED setting. In a recent large prospective observational study of patients with cancer presenting to the ED, Yilmaz et al. found that only half self-reported prior advance directives [1], which is similar to the rates of documented ACP in our study.

Clearly, despite the many initiatives instituted over the past decade to improve ACP in patients with cancer, much work remains to align patients’ quality-of-life goals with the reality of CPR outcomes. Our objective data on CPR outcomes and patients’ transition to DNR status offer powerful tools for counseling patients on their chances for meaningful recovery. ED providers play a crucial role in guiding families through difficult decisions about CPR, which may only prolong the dying process and cause unnecessary distress in advanced cancer patients. Too often these conversations occur after arrest in the ED, missing opportunities to have these dialogues with terminally ill patients before their inevitable decline. Continued provider education in holding these difficult discussions is vital for managing expectations and ensuring that patients’ and families’ goals align with the realities of cardiopulmonary arrest outcomes that our data provided.

Further prospective research is needed to explore the question of whether prior ACP lowers the rate of CPR events, although we think it likely does. In our institution, the number of CPR-associated deaths in the ED did not show any trends over the 7 study years; however, the numbers of such deaths were too small to be conclusive. (Table 5) We hypothesized that prior ACP discussion would increase the number of patients choosing DNR status owing to poor prognosis, leading to CPR for patients with an increased likelihood of meaningful survival but we did not find a difference in mortality between those with and without prior GOC discussion (Appendix A). The number of avoided CPR attempts owing to GOC discussions and the reason for a change in DNR status after resuscitation for patients with prior ACP are beyond the scope of this retrospective paper but could be the subject of future research. Because the number of patients with cancer who arrest in the ED at any institution is likely small, systematic reviews with meta-analyses and multicenter prospective studies are the most viable strategies to answer these questions. Furthermore, exploring emotional distress and financial toxicity among family members would greatly enhance our understanding of the effects of ACP on resuscitation.

### Limitations

As with any single-center study, we may have had unaccountable variables unique to our institution and results that may not have wider applicability. For example, some of our patients who required resuscitation may have not been included in our study because they required transportation to closer hospitals while unstable. Additionally, we may not have captured all CPR events in our ED, as initial reporting was voluntary using a safety intelligence system, whereas reporting in the final 2 years was mandated and performed by members of a “Code Team”, dedicated to maintaining these statistics. Likewise, we may not have captured all discussions around goals of care, as they may have been buried within notes or they may not have been documented at all.

## 5. Conclusions

We found that cardiopulmonary resuscitation of cancer patients in the ED is less frequent than for the general population (0.55 per 1000 vs. 1.2 per 1000) [22]. Despite high ROSC rates (67%), most patients transition to DNR status (72%), the median survival duration is 26 h, and few of them survive to discharge (15%). Mortality and hospital costs are equal in patients with and without prior GOC discussions, but those with documented GOC conversations have statistically significant higher rates of post-cardiac arrest conversion to DNR and shorter hospital and ICU lengths of stay, perhaps because prior ACP discussion prepared patients and their families to opt for a palliative approach. The reason for the decrease in resuscitation rates for ED cancer patients and increased conversion to DNR for those with previously documented ACP requires further study. Nevertheless, our findings may assist providers in managing expectations and ensuring that patient and family goals align with the realities of cardiopulmonary arrest outcomes. These results may also be valuable in guiding policymaking and resource distribution within hospital systems. 

## Figures and Tables

**Figure 1 cancers-16-02835-f001:**
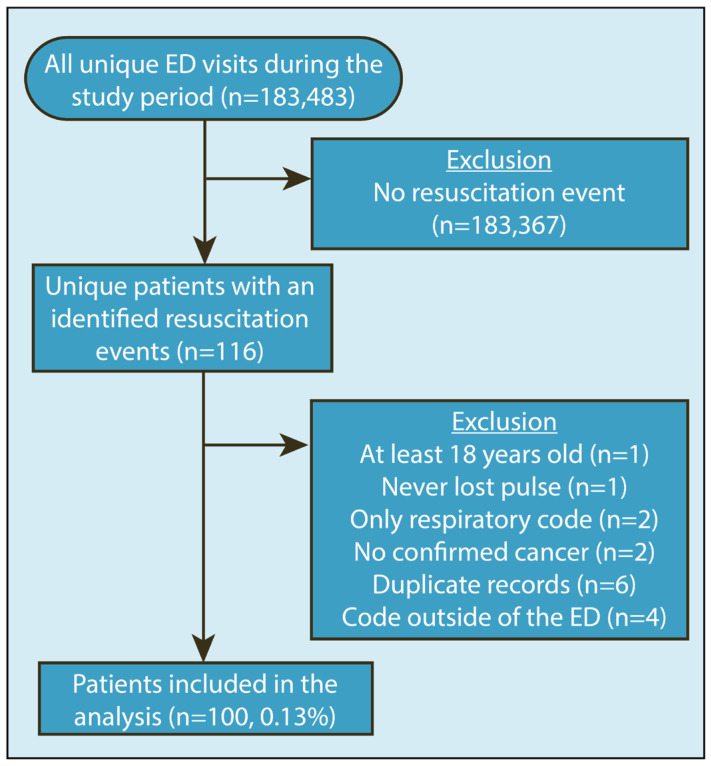
Study cohort flow diagram identifying patients with cancer who underwent resuscitation in the ED.

**Figure 2 cancers-16-02835-f002:**
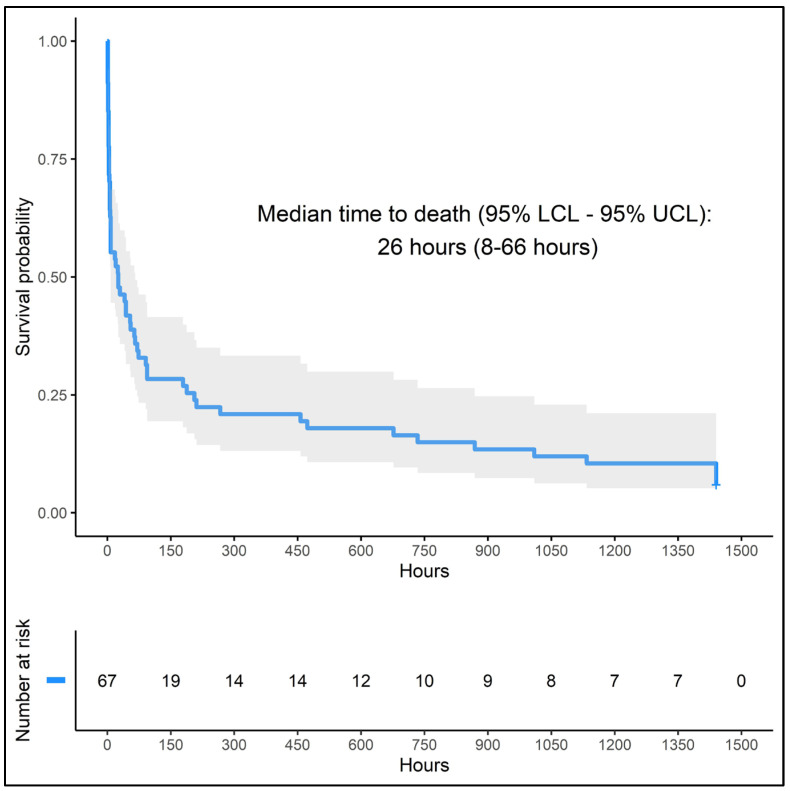
Median time to death for patients with cancer who had ROSC in the ED.

**Figure 3 cancers-16-02835-f003:**
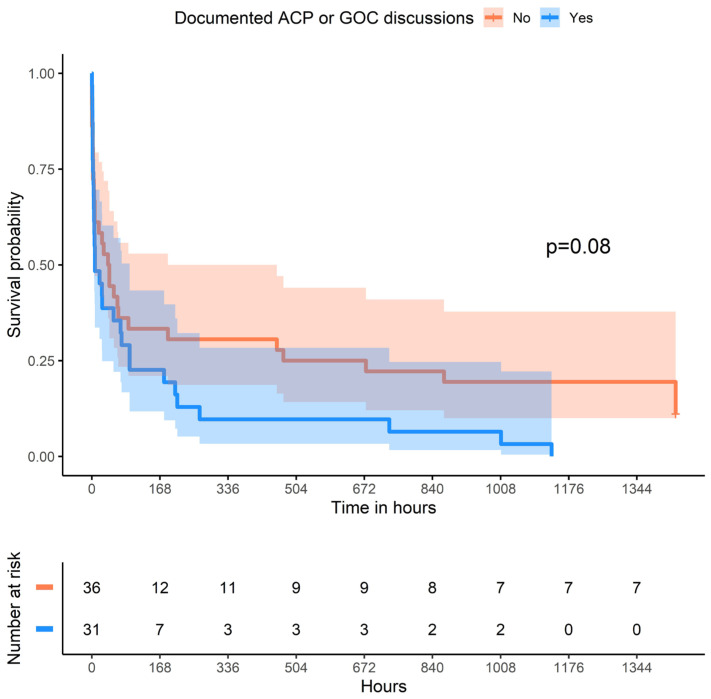
Kaplan–Meier survival curves for patients who achieved ROSC in the ED, with and without prior ACP or GOC documentation.

**Table 1 cancers-16-02835-t001:** Demographics and comorbidities for patients with cancer who underwent CPR in the ED (n = 100).

Characteristic	n (%)
Mean age, years (SD)	62 (12)
Sex	
Female	50 (50)
Male	50 (50)
Race	
Asian	5 (5)
Black or African American	19 (19)
White or Caucasian	63 (63)
Other *	13 (13)
Ethnicity	
Hispanic or Latino	16 (16)
Not Hispanic or Latino	84 (84)
Median CCI score [IQR]	6 [5–9]

Abbreviations: SD, standard deviation; CCI, Charlson comorbidity index; IQR, interquartile range. * Including American Indian, Alaska Native, Native Hawaiian, Other Pacific Islander, or multiracial.

**Table 2 cancers-16-02835-t002:** Cancer-related factors in patients with cancer who were resuscitated in the ED (n = 100).

Variable	n (%)
Cancer type	
Gastrointestinal	17 (17.0)
Lung	16 (16.0)
Leukemia	14 (14.0)
Head and neck	13 (13.0)
Other hematologic *	11 (11.0)
Genitourinary	9 (9.0)
Gynecologic	7 (7.0)
Breast	4 (4.0)
Sarcoma	3 (3.0)
Brain and spinal cord	2 (2.0)
Melanoma and other skin	2 (2.0)
Endocrine	1 (1.0)
Other	1 (1.0)
Distant metastasis at the time of presentation ^†^	
No	25 (34.2)
Yes	48 (65.8)
Received cancer therapy within 2 months before CPR event	
No	15 (15.0)
Yes	85 (85.0)

* Hematologic cancers besides leukemia. ^†^ Excluding patients with hematologic or central nervous system cancers (n = 73).

**Table 3 cancers-16-02835-t003:** Outcomes and code status for patients with cancer who underwent CPR in the ED (n = 100).

Variable	n (%)
ROSC achieved	67 (67.0)
Time to ROSC *, minutes [IQR]	11 [5.5–20.0]
Mortality rates	
24 h	65 (65.0)
48 h	72 (72.0)
72 h	77 (77.0)
7 days	81 (81.0)
14 days	86 (86.0)
30 days	89 (89.0)
60 days	93 (93.0)
90 days	94 (94.0)
In-hospital	85 (85.0)
ACP note or GOC documentation before the event	
No	56 (56.0)
Yes	44 (44.0)
DNR order revoked prior to CPRevent	6 (6.0)

* Only for patients who achieved ROSC.

**Table 4 cancers-16-02835-t004:** ED visits, resuscitation events, ROSC outcomes and hospital charges per year.

Patient Description	Year of ED Encounter Visit							
	2016 *	2017	2018	2019	2020	2021	2022	Total
All patient visits, N	21,933	26,861	27,197	28,305	21,982	27,337	29,868	183,483
Resuscitation events, N (%)	8 (0.036)	8 (0.030)	15 (0.055)	12 (0.042)	15 (0.068)	16 (0.059)	26 (0.087)	100 (0.055)
Achieved ROSC, N (%)	8 (100%)	6 (75%)	13 (87%)	6 (50%)	12 (80%)	11 (69%)	14 (54%)	70 ^#^ (70%)
Total hospital stay cost ($)	684,164	1,190,083	1,261,087	220,993	1,849,740	1,595,168	864,458	7,665,693

* Patient data for 2016 began on 1 March 2016. ^#^ Higher than in Table because some patients arrested more than once.

**Table 5 cancers-16-02835-t005:** Demographics and clinical characteristics stratified by ACP or GOC documentation prior to the resuscitation event for the patients who achieved ROSC (n = 67).

Variable	ACP or GOC Documentation before the Event		*p*
	No (n = 36)	Yes (n = 31)	
Age in years, mean ± SD	60.75 ± 13.65	60.23 ± 10.18	0.861
Sex			0.169
Female	16 (44.4)	19 (61.3)	
Male	20 (55.6)	12 (38.7)	
Race			0.826
Non-White	13 (36.1)	12 (38.7)	
White	23 (63.9)	19 (61.3)	
Ethnicity			0.547
Hispanic or Latino	5 (13.9)	6 (19.4)	
Neither Hispanic nor Latino	31 (86.1)	25 (80.6)	
CCI score, median [IQR]	6 [5–7]	8 [6–11]	0.013
Distant metastasis at the time of presentation *			0.382
No	11 (40.7)	6 (28.6)	
Yes	16 (59.3)	15 (71.4)	
Active cancer therapy			0.471
No	7 (19.4)	4 (12.9)	
Yes	29 (80.6)	27 (87.1)	
Code status changed after resuscitation			
No	14 (38.9)	5 (16.1)	0.039
Yes	22 (61.1)	26 (83.9)	
Time to ROSC ^†^ in minutes, median [IQR]	14 [6–21]	10 [6–19]	0.525
In-hospital mortality			0.529
No	8 (22.2)	5 (16.1)	
Yes	28 (77.8)	26 (83.9)	
ICU length of stay in days, median [IQR]	2.25 [0.73–13.54]	0.92 [0.29–6.88]	0.028
Hospital length of stay in days, median [IQR]	2.60 [0.94–17.29]	1.17 [0.42–7.96]	0.046
Total cost of hospital stay in US$, median [IQR]	50,567 [20,951–213,103]	44,402 [25,795–134,974]	0.904
Time to death in hours, median [IQR]	42 [7–188]	8 [5–93]	0.080 ^a^

Data reported as n (%) unless specified otherwise. * Only for patients with solid tumors. ^†^ Only for patients who achieved ROSC. ^a^ Log-rank test.

## Data Availability

The data presented in this study are available upon request from the corresponding author. The data are not publicly available, and IRB approval from MD Anderson Cancer Center is required to share the data.

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
