# Peer review of "Prior Advanced Care Planning and Outcomes of Cardiopulmonary Resuscitation in the Emergency Department of a Comprehensive Cancer Center"

_cancers, 2024, doi:10.3390/cancers16162835_

Round 1

Reviewer 1 Report

Comments and Suggestions for Authors

Congratulations!

General  comments:

Thank you for giving me the opportunity to review the paper entitled:

Prior Advanced Care Planning and Outcomes of Cardiopulmonary Resuscitation in the Emergency Department of a Comprehensive Cancer Center”.

I appreciate, it was real pleasure and an honour.

The number of cancer patients is rapidly growing so the need for treatment results analysis increases as well. The subject of the study that is prior advanced care planning (ACP) in regards to need for cardiopulmonary resuscitation (CPR) in emergency department constitutes an important clinical problem and the challenge so the presented for review paper is extremely important.

Specific  comments:

The observation period (6,5 years) and the study group, n=100 of 183,483 (0.05%) in ED are sufficient enough to conclude important remarks and recommendations.

The preseted study is a single center experience conducted in the hospital aware of  advanced care planning need. Even though 56% of the pts were unable to specify their  preferences immediately before CPR.  Only six patients revoked their prior DNR orders ! It can be expected what might be the results if it is multicenter study. Distant metastases were present in 66% of the patients, very few (15%) survived to discharge.

These statements: ” In general, there is a lack of adequate advanced care planning in cancer patients presenting to the ED, despite the well-documented need for establishing goals of care (GOC) before a stressful and urgent medical crisis” are crucial conclusions and should be widely posted in medical literature.Despite the many initiatives instituted over the past decade to improve ACP in patients with cancer, much work remains to align patients’ quality-of-life goals with the reality of CPR outcomes”.

In my opinion it would be usufull for the manuscript to include some relevant information for educational purpose for the readers, let’s hope wide medical community, concerning treatment options and care towards the end of life, for example:

The most challenging decisions in this area are about withdrawing or not starting a treatment such as CPR because in certain circumstances this may only prolong the dying process or cause unnecessary distress. The physicians involved in cancer patient’s advanced care planning  should be trained  in the decision-making process in a hospital setting. The physician must  record these decisions made about a patient’s treatment and care including CPR and DNACPR orders.

I truly recommend this paper for publication in in the present form after just a minor revision, with few sentences added in discussion section mentioned above, if the authors agree to do so. The presented study is easy to read, interesting and extremely helpful in discussion concerning end of life care. The presented data on CPR outcomes and patients’ transition to DNR status offers powerful tools for counseling patients and their famillies on the chances for meaningful recovery with good quality of life.

Author Response

Reviewer 1

Congratulations!

General  comments:

Thank you for giving me the opportunity to review the paper entitled:

Prior Advanced Care Planning and Outcomes of Cardiopulmonary Resuscitation in the Emergency Department of a Comprehensive Cancer Center”.

I appreciate, it was real pleasure and an honour.

The number of cancer patients is rapidly growing so the need for treatment results analysis increases as well. The subject of the study that is prior advanced care planning (ACP) in regards to need for cardiopulmonary resuscitation (CPR) in emergency department constitutes an important clinical problem and the challenge so the presented for review paper is extremely important.

Specific  comments:

Thank you for your thorough review and salient observations and comments. Your positive feedback and constructive comments were invaluable in shaping the final version of the manuscript and we have tried our best to address all the questions raised.

Comment 1: The observation period (6,5 years) and the study group, n=100 of 183,483 (0.05%) in ED are sufficient enough to conclude important remarks and recommendations.

Thank you.

Comment 2: The preseted study is a single center experience conducted in the hospital aware of  advanced care planning need. Even though 56% of the pts were unable to specify their  preferences immediately before CPR.  Only six patients revoked their prior DNR orders ! It can be expected what might be the results if it is multicenter study. Distant metastases were present in 66% of the patients, very few (15%) survived to discharge.

Yes, we would love to further this research with a multicenter study in the future.

Comment 3:  These statements: ” In general, there is a lack of adequate advanced care planning in cancer patients presenting to the ED, despite the well-documented need for establishing goals of care (GOC) before a stressful and urgent medical crisis” are crucial conclusions and should be widely posted in medical literature. „Despite the many initiatives instituted over the past decade to improve ACP in patients with cancer, much work remains to align patients’ quality-of-life goals with the reality of CPR outcomes”.

We hope we have highlighted these crucial conclusions throughout the manuscript.

Comment 4: In my opinion it would be usufull for the manuscript to include some relevant information for educational purpose for the readers, let’s hope wide medical community, concerning treatment options and care towards the end of life, for example:

The most challenging decisions in this area are about withdrawing or not starting a treatment such as CPR because in certain circumstances this may only prolong the dying process or cause unnecessary distress. The physicians involved in cancer patient’s advanced care planning  should be trained  in the decision-making process in a hospital setting. The physician must  record these decisions made about a patient’s treatment and care including CPR and DNACPR orders.

We have discussed the national and institutional initiatives around early advanced care planning and enhanced documentation of these discussions. The conclusion and final discussion mentions how we hope our research will be used to inform decision making and policies.

Based on your comments I have added the italicized text.

Our objective data on CPR outcomes and patients’ transition to DNR status offers powerful tools for counseling patients on their chances for meaningful recovery. ED providers play a crucial role in guiding families through difficult decisions about CPR, which may only prolong the dying process and cause unnecessary distress in advanced cancer patients. Too often these conversations occur after arrest in the ED, missing opportunities to have these dialogues with terminally ill patients before their inevitable decline. Continued provider education in holding these difficult discussions is vital for managing expectations and ensuring that patient’s and family’s goals align with the realities of cardiopulmonary arrest outcomes that our data has provided.

Comment 5: I truly recommend this paper for publication in in the present form after just a minor revision, with few sentences added in discussion section mentioned above, if the authors agree to do so. The presented study is easy to read, interesting and extremely helpful in discussion concerning end of life care. The presented data on CPR outcomes and patients’ transition to DNR status offers powerful tools for counseling patients and their famillies on the chances for meaningful recovery with good quality of life.

Thank you again for supporting our paper.  We are very honored.

Reviewer 2 Report

Comments and Suggestions for Authors

It is a research paper that describes CPR in patients with cancer in emergency care, especially those with advanced care planning (ACP). It is a very interesting issue with wise conclusion. It is retrospective with a small number of patients. However, its scientific value is great. I have some remarks and the authors need to answer:

1. The introduction seems verbalistic enough and needs to be limited.

2. More data need to statistical analysis in a separate paragraph. For example how the normality of continuous variables was estimated. Which variables had not a normal variance and no parametric test were needed.

3. The authors need very well the limitations of their study and describe them in a separate paragraph.

4Finally the conclusion needs to be limited in 1-2 sentences.

Author Response

Reviewer 2

It is a research paper that describes CPR in patients with cancer in emergency care, especially those with advanced care planning (ACP). It is a very interesting issue with wise conclusion. It is retrospective with a small number of patients. However, its scientific value is great. I have some remarks and the authors need to answer:

We sincerely appreciate the reviewer’s time and effort in evaluating our paper and we have made every effort to address all of the questions raised.

Comment 1:  The introduction seems verbalistic enough and needs to be limited.

We have edited down the introduction by over 100 words, as documented by red markings below. We hope this makes it less verbose will providing the essential information.

Introduction

  As the incidence and survival rates for cancer continue to rise worldwide, the number of cancer patients presenting to the emergency department (ED) also has also increased. Whereas morbidity and mortality rates have improved for many patients, those needing emergency care often have advanced disease, and a palliative approach may be more appropriate than aggressive treatment [1]. Discriminating those patients likely to benefit from aggressive resuscitation from those whose rapid demise is inevitable who won’t is becoming increasingly important. The urgency in providing life-saving critical care In the ED, cancer patients with incomplete or unknown histories will often present often without a detailed history, can requiring urgent life-saving interventions which limits goal setting discussions, leading to inappropriate or unintended decisions regarding resuscitation [2].

  Successful cardiopulmonary resuscitation (CPR) with meaningful recovery is limited in the general population (with one meta-analysis by Zhu from 2015 demonstrating a rate of 15%), and these with similar rates are not markedly worse for the cancer subgroups (2%-18%) [3-7]. Despite these lackluster rates, large cardiovascular care organizations are reluctant to endorse any pre-cardiac arrest prediction tools to identify patients who are unlikely to survive with favorable neurologic outcomes and to help guide goals of care discussions [8].

  In the decade since the Zhu data, there has been increasing emphasis on establishing realistic goals of care for patients with advanced cancer and prioritizing their quality of life. Cost containment and resource constraints, such as those experienced during the COVID-19 pandemic, have accelerated this agenda and influenced outcomes [9-11]. For example, the rise in seriously ill patients during the COVID-19 pandemic led to a concurrent rise in palliative care offerings, even in EDs [12]. In our dedicated cancer care institution, Advanced care planning (ACP) became more prevalent during this period without evidence of a decrease in overall survival[9,10,13] In general, there is a lack of adequate Nevertheless, advanced care planning in many cancer patients presenting to the ED remains inadequate, despite the well-documented need for establishing goals of care (GOC) before a stressful and an urgent medical crisis [1,14-16]. One study demonstrated that only 24% of hospitalized cancer patients discussed end-of-life care with any provider despite 82.5% of these patients wanting to do so [16].

  To address this gap in care, Leaders in both emergency medicine and palliative care medicine havecalled for the introduction of palliative care in EDs [17,18]. In 2015, the National Institutes of Health created an oncology-specific emergency and urgent care consortium known as the Comprehensive Oncologic Emergencies Research Network (CONCERN), which identified ED based palliative care a priority in the emergency setting as a prioritized research area [19]. From 2020 through 2023, an alliance of 10 dedicated cancer centers in the United States issued a call to improve goal-concordant care in keeping with rooted in national quality initiatives for seriously ill patients, such as . These centers recommended the implementation of formal communication skills training, structured electronic record ACP, mandated GOC conversations and tracking metrics. GOC documentation in the electronic medical record, expectations of GOC conversations taking place, and tracking metrics for these goals [20].

  Over the past 4 years, our institution responded to these recommendations and the COVID-19 epidemic by launching a Goals of Care Project to identify populations at high risk for decompensation and establish GOC early in their the patient’s treatment trajectory [10]. The project resulted in decreased ICU admissions, hospital length of stay and hospital mortality, as well as alongside improved patient symptom scores and performance status [9,10]. However, how, and to what extent Yet, how this focus on palliative care impacted the frequency and outcomes of cancer patient resuscitation in the ED has not been yet to be studied.

  The aim of our study was to update the data on CPR outcomes for cancer patients presenting to the ED and to explore the statistical association of prior GOC discussion or ACP with the post-resuscitation course for patients who achieved ROSC. Our study focuses on patients in the ED, where the urgency of resuscitation may supersede obtaining a full medical history or engaging with the patient in discussions of prognosis, unlike the hospital ward or ICU. Some of our patients made their resuscitation wishes known through prior directives or discussions in the ED, but others never had the opportunity to express their preference prior to CPR.  We hypothesized that the patients with who had prior goals of care discussions and were willing to undergo resuscitation did so because they had a more favorable cancer prognosis and therefore would have expected we would expect better outcomes.  

Comment 2: More data need to statistical analysis in a separate paragraph. For example how the normality of continuous variables was estimated. Which variables had not a normal variance and no parametric test were needed.

As requested, we have now added more explanation on how we tested normality for the continuous variables and which variables met the normality assumption for which we used parametric tests.

Methods section: The normality of the continuous variables was assessed using histograms, box plots, Q-Q plots, and the Shapiro-Wilk test, for which only age met the normality assumption.

Comment 3:  The authors need very well the limitations of their study and describe them in a separate paragraph.

Thank you for your kind assessment.

Comment 4: Finally the conclusion needs to be limited in 1-2 sentences.

Though we understand that a very limited conclusion can be very impactful, we could not fit all of our critical findings and conclusions in just 2 sentances, and believe the summation of our results is important for those readers who focus on the conclusion. Also, in reviewing other Cancers articles, we found many had 1-2 paragraph conclusions.  Nevertheless, we significantly shortened the conclusion as shown below.

We found that cardiopulmonary resuscitation of cancer patients in the ED is less frequent than for the general population (0.55 per thousand vs. 1.2 per thousand) [22]. Despite high ROSC rates (67%), most patients transition to DNR status (72%), median survival duration is 26 hours, and few of them survive to discharge (15%). Mortality and hospital costs are equal in patients with and without prior GOC discussions, but those with documented GOC conversations have statistically significant higher rates of post-cardiac arrest conversion to DNR and shorter hospital and ICU lengths of stay, perhaps because ACP discussion prepared patients and their families to opt for a palliative approach. The reason for the decrease in resuscitation rates for ED cancer patients and increased conversion to DNR for those with previously documented ACP requires further study. Nevertheless, our findings may assist providers in managing expectations and ensuring that patient and family goals align with the realities of cardiopulmonary arrest outcomes. These results may also be valuable in guiding policymaking and resource distribution within hospital systems.